# A Load-Independent Current/Voltage IPT Charger with Secondary Side-Controlled Hybrid-Compensated Topology for Electric Vehicles

Guangyao Li, Cheol-Hee Jo, Chang-Su Shin, Seungjin Jo and Dong-Hee Kim *

Department of Electrical Engineering, Chonnam National University, 77, Yongbong-ro, Buk-gu, Gwangju 61186, Korea
* Correspondence: kimdonghee@jnu.ac.kr; Tel.: +82-62-530-1736

**Abstract:** The inductive power transfer (IPT) method is an emerging charging technology that has some advantages over traditional plug-in systems. For example, it is safer, more convenient, and efficient, leading to its widespread acceptance. To design an IPT charger capable of providing a load-independent output, this paper proposes a secondary side-controlled hybrid-compensated topology used in the IPT system to charge the battery with a constant current/voltage output. According to an analysis of the Π-type network, effectively using the existing configuration compensation parameters and adding two AC switches to perform hybrid-topology switching reduces the system's passive components. Additionally, the proposed IPT charger can easily realize zero-voltage switching. The secondary side-based control omits wireless communication links. Moreover, the control strategy is relatively simple, enhancing the system's reliability. We designed a 1.4 kW experimental prototype with a 15 cm air gap between the transmitter and receiver to verify the proposed hybrid-compensated IPT system's feasibility.

**Keywords:** inductive power transfer; hybrid-compensated topology; constant current/voltage output; battery charging

## 1. Introduction

The emergence of inductive power transfer (IPT) technology using magnetic-field coupling has provided a feasible alternative to battery charging based on physical connections. Owing to the advantages of isolation, high transfer efficiencies, and convenience, IPT has potential for various battery-charging applications, such as low-power portable electronic devices [1], high-power electric vehicles (EVs) [2,3], and automated guided vehicles [4]. Additionally, IPT technology could be applied to charge underwater equipment and biomedical implants, in addition to improving the vehicle-to-grid connection success rate.

As shown in Figure 1, a typical IPT system consists of a power supply, a power factor correction (PFC) converter, an inverter, a compensation resonant network, a loosely coupled transformer (LCT), a rectifier, and a battery load. The modules in the IPT system have different functions. The PFC converter is mainly responsible for providing different levels of DC-link voltage. The inverter converts DC into high-frequency AC and transmits it to the resonant network on the primary side. The AC generates a high-frequency magnetic field through the transmitting coil and transmits it to the aligned receiver coil through a large air gap with the primary and secondary side compensation networks, which are used to decrease each coil's reactive current. In the coupled magnetic field, the alignment of the coils directly affects the system's performance. Tan et al. [5] proposed a mesh-based coil precise positioning strategy using the voltage principle induced by the detection coil installed on the secondary coil. This system could be applied in automatic parking scenarios

to help align the coils and improve electrical transmission performance. Finally, the battery is charged after rectification and filtering.

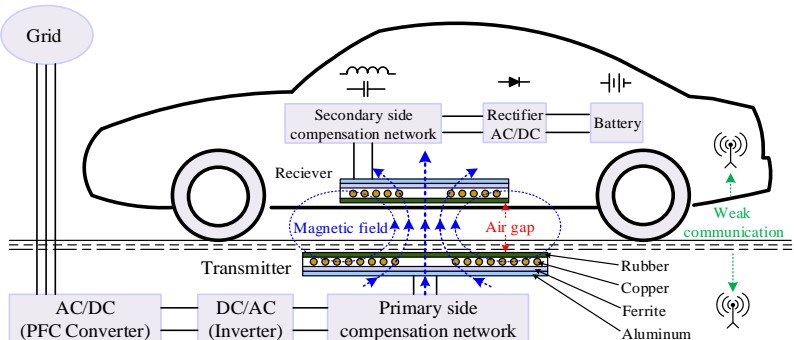

**Figure 1.** A typical IPT EV charging system.

Generally, the battery's main charging modes are the constant current (CC) and constant voltage (CV) modes [6], which form the classic charging curves shown in Figure 2. As shown, the battery is charged by an initial constant charging current $I_b$. However, to avoid battery overcharge, when the charging voltage reaches the preset threshold voltage $U_b$, the IPT charger switches to the CV mode, which helps extend battery life.

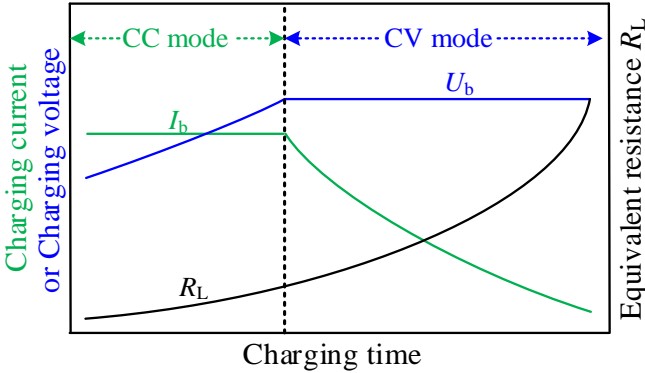

**Figure 2.** A battery's typical charging curve.

The battery's equivalent resistance exhibits a nonlinear trend, making it difficult for the IPT charging system to obtain high-efficiency CC/CV outputs. To design a stable, reliable, and load-independent IPT charger with CC and CV outputs, researchers have proposed four types of control schemes: frequency-modulation (FM) control, frequency-switching control, phase-shift (PS) control, and additional DC/DC converters. Liu et al. [7] proposed an IPT system with high-efficiency CC/CV output capabilities based on the FM control method. However, Zhao et al. [8] pointed out that the frequency-division phenomenon caused by the wide range of frequency changes detunes the system. Additionally, the large amount of generated reactive power reduces the system's efficiency. Moreover, Wang et al. [9] analyzed that the frequency bifurcation phenomenon can cause stability problems; however, if a controllable operation is achieved in the bifurcation region, the system's power transfer capability could be significantly improved. Schormans et al. [10] proposed an optimal frequency tracking method to ensure that the link operates at the resonant point and maximizes the output voltage when frequency splitting phenomena occur around the resonant frequency of 5 MHz. Vu et al. [11] proposed a frequency-switching control scheme that can realize the CC/CV charging mode under the ZPA condition without causing a frequency-division phenomenon and without any additional switches. However, this method is realized in a wide frequency range, increasing the design difficulty and reducing its freedom because the CV mode's output characteristics are limited

by the design of the compensation parameters in the CC mode. The third scheme can obtain CC and CV outputs that are independent of the load by using the PS control [12,13]. However, for a wide range of PSs, the power switch's zero-voltage switching (ZVS) is not easily achievable, thus increasing the volt-ampere (VA) rating and the switching loss. However, these methods are complex and need real-time wireless communication links between the transmitter and receiver. In order to overcome the limitations of such control schemes, Li et al. [14] and Huang et al. [15] developed a novel one by adding DC/DC converters on both sides of a traditional IPT system to control output by modifying the duty cycle. However, the increases in weight, volume, and system power loss caused by the DC/DC converter cannot satisfy the receiver's high-power density. Moreover, the foregoing control methods increase the difficulty of the control loop design.

Considering these problems, another scheme that uses AC switches (ACSs) to promptly switch the compensated topology and obtain load-independent CC/CV outputs has been widely accepted because of its simple control scheme and low cost. The reconfigured compensated topology can be considered as a combo of different forms of passive resonance networks [16]. This scheme mainly includes the combination of low- or high-order hybrid-compensated topologies. The low-order hybrid topology mainly realizes load-independent CC and CV outputs by using different combinations of the four basic compensation topologies, i.e., series–series (S–S), parallel–parallel (P–P), series–parallel (S–P), and parallel–series (P–S) [17]. For realizing ZVS, the S–S and P–P topologies can charge the battery in the CC mode, while the S–P and P–S topologies can charge the battery in the CV mode under a dynamic load [18–20]. According to the four types of basic compensated topologies, Auvigne et al. [21] proposed an IPT system with CC and CV outputs based on S–S and S–P topologies. Qu et al. [22] achieved two CC/CV outputs by switching between S–S and P–S topologies. However, the limitations of the LCT parameters limit the design freedom of low-order IPT systems. To prevent the IPT system's output from being restricted by the LCT parameters, researchers have proposed a higher-order compensation topology composed of multiple compensation components. For example, the double-sided LCC (DS-LCC) and LCC-S compensated topologies utilize multiple interactive resonances of multiple compensation capacitors and inductors to implement CC/CV output characteristics, respectively [23–30]. Higher-order topologies can achieve different levels of CC or CV outputs by adjusting the compensation inductance without changing the LCT. Therefore, Li et al. [31] proposed an IPT charger based on a combination of LCC-S and DS-LCC topologies to achieve CC/CV outputs while simultaneously achieving low reactive power and ZVS. Chen et al. [32] proposed a variable-parameter IPT system with CC/CV outputs. Although only one ACS is used, a compensated capacitor is added, and a zero-phase angle (ZPA) is not implemented in the CC mode. An analysis of these models has indicated that the combination of low- and high-order compensation topologies usually adds additional reactive components, increases the weight, volume, and cost, and necessitates real-time wireless communication links. Moreover, to achieve CC and CV outputs, a complex feedback control link is typically added, increasing design difficulty. Therefore, a well-designed compensated network has significant advantages for realizing CC or CV outputs under open-loop conditions. However, ideas based on the secondary side involving the adoption of hybrid-compensated high-order topologies with near-communication have not been sufficiently investigated for realizing the CC/CV modes.

In keeping with the foregoing discussion, this paper proposes a switching hybrid-compensated topology for the battery CC/CV charging mode on the secondary side. Only two ACSs are utilized on the transmitter to achieve the CC/CV mode. This study's main contributions are as follows: (1) Using the same drive signal to control the ACSs, the control strategy is relatively simple. (2) Compared with the conventional DS-LCC topologies hybrid switching method, only two ACSs need to be added on the secondary side, implying that no additional compensation devices are needed. Thus, the proposed system avoids unnecessary increases in the secondary side's volume and weight. (3) Secondary-side control is adopted, and wireless communication links for CC/CV charging control are not

required, which ensures the proposed IPT system's robustness and reliability. (4) A systematic analysis of the proposed IPT systems was conducted. Finally, a 1.4 kW experimental prototype was established to validate it. Measurement results indicated that the system's maximum efficiency in the CC and CV modes was about 92.2% and 92.3%, respectively.

The remainder of this paper is organized as follows: Section 2 presents a theoretical analysis of the proposed IPT systems. In Section 3, a hybrid compensation IPT system that can realize CC/CV outputs is proposed. Section 4 describes the experimental procedure used to validate the proposed system. We present our conclusions in Section 5.

## 2. Theoretical Analysis and Proposed Hybrid IPT System

### 2.1. The Principle of Constant Voltage/Current Outputs of Second- and Third-Order Resonant Networks

In the IPT system, the LCT's coupling coefficient $k$ is small, resulting in a large amount of inductive reactive power. To reduce it, a compensation network must be added. Meanwhile, CC or CV output characteristics independent of the output impedance can be realized by using the passive resonant network. Figure 3a shows a $\Gamma$-style resonant network with a CV output driven by a current source, and Figure 3b shows a $T$-style resonant network with a CC output driven by a voltage source.

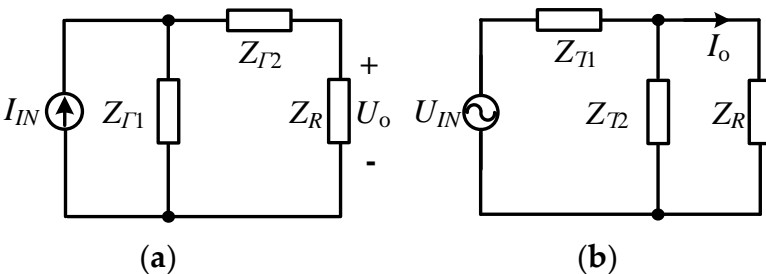

(a)                                                                                  (b)

**Figure 3.** (**a**) A $\Gamma$-style resonant network; (**b**) A $T$-style resonant network.

In Figure 3a, $I_{\text{IN}}$ represents a CC source; $Z_{\Gamma 1}$ and $Z_{\Gamma 2}$ represent the impedances of the compensation components; $Z_R$ represents the load impedance; the load voltage $U_\text{o}$ and input impedance $Z_{\Gamma \text{IN}}$ are expressed as follows:

$$\begin{cases} U_o = \dfrac{Z_{\Gamma 1} Z_R I_{IN}}{Z_{\Gamma 1} + Z_{\Gamma 2} + Z_R} \\ Z_{\Gamma IN} = (Z_R + Z_{\Gamma 2}) || Z_{\Gamma 2} = X^2/Z_R - Z_{\Gamma 1} \end{cases} \tag{1}$$

where $Z_{\Gamma 1} = jX_{\Gamma 1}$ and $Z_{\Gamma 2} = -jX_{\Gamma 2}$. To achieve a CV output, $Z_{\Gamma 1} + Z_{\Gamma 2}$ should be equal to 0, and the output voltage can be represented as follows: $U_o = Z_{\Gamma 1} I_{IN} = -Z_{\Gamma 2} I_{IN}$. Additionally, the output voltage should be independent of the dynamic loads. Thus, the CC−source input should include the impedance $Z_{\Gamma 1}$ to neutralize the impedance in the $\Gamma$ resonant network. The $\Gamma$-style resonant network has a CV output.

In Figure 3b, $U_{\text{IN}}$ represents a CV source, and $Z_{T1}$ and $Z_{T2}$ represent the impedances of the resonant components. The output current $I_o$ and input impedance $Z_{T\text{IN}}$ can be represented as follows:

$$\begin{cases} I_o = \dfrac{Z_{T2} U_{IN}}{Z_{T1} Z_{T2} + (Z_{T1} + Z_{T2}) Z_R} \\ Z_{TIN} = Z_R // (-jX) + jX = X^2/(Z_R - jX) \end{cases} \tag{2}$$

where $Z_{T1} = jX_{T1}$ and $Z_{T2} = -jX_{T2}$. Clearly, to satisfy the load-independent purely resistive CC output, only when $Z_{T1} + Z_{T2} = 0$ and $Z_R$ contains $Z_{T1}$, the output current $I_o$ can be expressed as: $I_o = U_{IN}/Z_{T1} = -U_{IN}/Z_{T2}$.

In addition to the second-order resonant network, by combining the $\Gamma$-style and $T$-style resonant networks, we can obtain a third$-$order $\Pi$-style resonant network as shown in Figure 4.

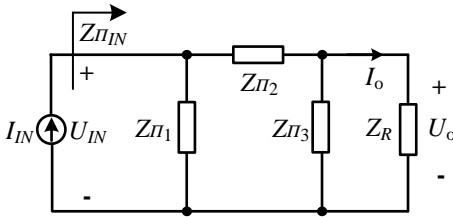

**Figure 4.** A $\Pi$-style resonant network.

According to Kirchhoff's current law, the $\Pi$-style resonant network can be described as:

$$I_o = \frac{Z_{\Pi 1}Z_{\Pi 3}I_{IN}}{(Z_{\Pi 1} + Z_{var\Pi 2})Z_{\Pi 3} + (Z_{\Pi 1} + Z_{\Pi 2} + Z_{\Pi 3})Z_R} \tag{3}$$

Hence, the transconductance $G_{II}$, which is defined as the ratio of the output current $I_o$ of the rectifier to the input current $I_{IN}$ of the high-frequency inverter (HFI), can be determined as:

$$G_{II} = \left| \frac{I_o}{I_{IN}} \right| = \left| \frac{Z_{\Pi 1}Z_{\Pi 3}}{(Z_{\Pi 1} + Z_{\Pi 2})Z_{\Pi 3} + (Z_{\Pi 1} + Z_{\Pi 2} + Z_{\Pi 3})Z_R} \right| \tag{4}$$

When the three resonant components in the $\Pi-$style resonant network satisfy the resonant condition $Z_{\Pi 1} + Z_{\Pi 2} + Z_{\Pi 3} = 0$, a CC output independent of the dynamic loads can be expressed:

$$G_{II} = \frac{Z_{\Pi 1}}{Z_{\Pi 1} + Z_{\Pi 2}} = -\frac{Z_{\Pi 1}}{Z_{\Pi 3}} \tag{5}$$

Similarly, the voltage transfer gain $G_{IV}$, i.e., the ratio of the output voltage $U_o$ to the input current $I_{IN}$, and the input transfer impedance $Z_{\Pi IN}$ are expressed as follows:

$$\begin{cases} G_{IV} = \left| \frac{U_o}{I_{IN}} \right| = \left| \frac{Z_{\Pi 1}Z_{\Pi 3}Z_R}{(Z_{\Pi 1} + Z_{\Pi 2})Z_{\Pi 3} + (Z_{\Pi 1} + Z_{\Pi 2} + Z_{\Pi 3})Z_R} \right| \\ Z_{\Pi IN} = \left| \frac{U_{IN}}{I_{IN}} \right| = \left| \frac{(Z_{\Pi 1}Z_{\Pi 3})^2 Z_R}{[(Z_{\Pi 1} + Z_{\Pi 2})Z_{\Pi 3} + (Z_{\Pi 1} + Z_{\Pi 2} + Z_{\Pi 3})Z_R]^2} \right| \end{cases} \tag{6}$$

In accordance with (6), the only way to let the $\Pi-$circuit have a CV output is to set $Z_{\Pi 3}$ to be infinite and let $Z_{\Pi 1} + Z_{\Pi 2} = 0$. Thus, the new voltage transfer gain $G_{IV}$ and $Z_{\Pi IN}$ are expressed as follows:

$$\begin{cases} G_{IV} = \left| \frac{U_o}{I_{IN}} \right| = \left| Z_{\Pi 1} \right| \\ Z_{\Pi IN} = \left| \frac{U_{IN}}{I_{IN}} \right| = \left| \frac{Z_{\Pi 1}^2}{Z_R} \right| \end{cases} \tag{7}$$

Therefore, the $\Pi-$circuit driven by the CC source can switch between the load-independent CC and CV output mode through the variable $Z_{\Pi 3}$. If the ACSs are set on the primary side, a wireless communication link is required. Thus, mode switching is expected to occur on the secondary side.

### 2.2. The Proposed Hybrid Compensation IPT System

This study's objectives are to use the control of the $\Pi$-style resonant network arranged in the receiver side to omit the wireless communication links and design a hybrid compensation topology that does not require additional compensation components and complex control logic. We propose a hybrid$-$compensated IPT system that combines the charac-

teristics of the $\Pi$-circuit and the DS$-$LCC compensated topology, in addition to realizing CC/CV charging modes regardless of load variation.

The proposed IPT system's circuit diagram is shown in Figure 5. The full-bridge HFI converts the DC voltage source $U_{DC}$ into a square-wave voltage source $U_{AB}$ through MOSFETs ($Q_1$–$Q_4$). The AC generated by the electromagnetic induction is rectified by a rectifier consisting of four diodes ($D_1$–$D_4$) and converted into DC through the filter capacitor $C_o$ for battery charging. As shown in Figure 1, the LCT's main components include copper wire coils, ferrite, and aluminum shielding plates. The self-inductors $L_p$, $L_s$ and the resonant components $L_1$, $L_2$, $C_{p1}$, $C_{p2}$, $C_{f1}$, and $C_{f2}$ can realize the transformation of the magnetic and electric fields on the premise of satisfying specific resonance conditions. Among them, $C_{f2}$ can be decomposed into two parts, i.e., $C_{fcc}$ is adopted to realize the CC mode, and $C_{fcv}$ is adopted to realize the CV mode. Details regarding parameter settings are presented in Section 3. The transmitter's resonant network consists of $L_1$, $C_{p1}$, $C_{f1}$, and $L_p$, and that of the receiver consists of $L_2$, $C_{p2}$, $C_{f2}$, and $L_s$. $M$ represents the mutual inductance.

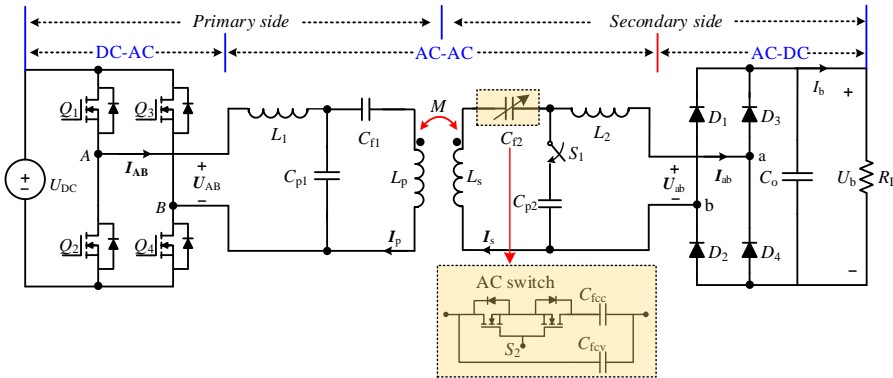

**Figure 5.** The proposed hybrid compensation topology's circuit diagram.

### 2.3. Analysis and Implementation of the CC Charging Mode with the ZPA Operation

Generally, for the analysis of an LCT composed of different combination forms, *M*-style and *T*-style models are used as shown in Figure 6. We consider that the theoretical analysis needs to utilize the *T*-type model to construct a $\Pi$-style model. In the proposed topology's CC mode, the compensation capacitor needs to be compensated by the LCT's leakage inductance to realize the ZPA adjustment. Meanwhile, the mutual inductance of the LCT and the secondary side should form a $\Pi$-type resonance network to achieve the CC output. Therefore, the *T*-style model is used in this study to analyze the proposed IPT systems.

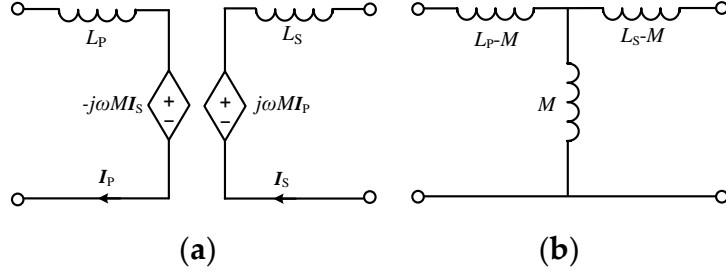

**Figure 6.** The equivalent model of the LCT: (**a**) $M-$type model; (**b**) $T-$type model.

As shown in Figure 5, the proposed IPT system works in the CC mode when both ACS $S_1$ and $S_2$ are set in the "on" state. Figure 7 presents the proposed IPT charger's equivalent circuit. The overall resonant compensation circuit can be divided into three parts: ① a $T-$style resonant network, ② the ZPA adjustment, and ③ a $\Pi$-style resonant network.

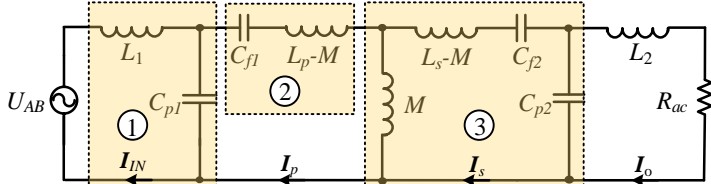

**Figure 7.** The proposed hybrid topology's equivalent analytical circuit for the CC mode.

According to the characteristics of the $T$-style resonance network, the input voltage source $U_{AB}$ cascading with $L_1$ and $C_{p1}$ is simplified to a current source $I_p$ as exhibited in Figure 8, which can be expressed as:

$$I_p = \frac{U_{AB}}{j\omega L_1} \tag{8}$$

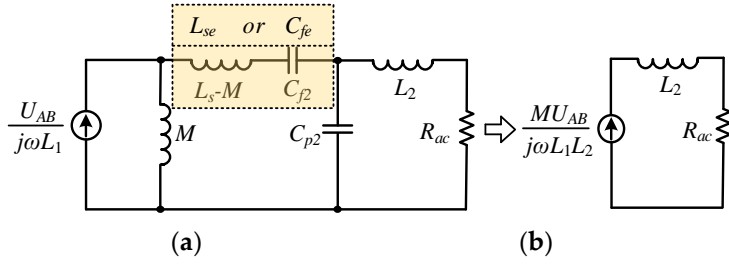

**Figure 8.** Circuit conversion process for the CC mode: (**a**) Circuit before conversion; (**b**) simplified circuit.

Here, $L_1$ and $C_{p1}$ should satisfy $\omega^2 = 1/(L_1 C_{p1})$. When $I_{IN}$ flowing in the transmitter coil works at the resonant frequency point, its magnitude is only affected by the input voltage on the transmitter side; it does not change with variations in the loads or the coupling coefficient $k$ on the secondary side. This helps form a uniform and stable magnetic induction intensity during the transmission process.

Then, after ② the ZPA adjustment, $I_p$ flows into the $\Pi-$style resonant network composed of $M$, $L_s$—$M$, $C_{f2}$, and $C_{p2}$. In practice, the leakage inductance $L_s$—$M$ and $C_{f2}$ can be regarded as one device, which may be an inductor $L_{se}$ or a capacitor $C_{pe}$. According to (5), when the sum of the impedances of the three resonant elements in the $\Pi-$style resonant network is equal to zero, a CC output independent of the load change can be obtained. The resonant condition of $M$, $C_{p2}$, $L_{se}$, and $C_{pe}$ can be given as follows:

$$\omega^2 = \frac{1}{MC_{p2}} + \frac{1}{MC_{pe}} \quad \text{or} \quad \omega^2 = \frac{1}{C_{p2}(M + L_{se})} \tag{9}$$

where $M = k/\sqrt{L_p L_s}$. Therefore, Figure 8a can be further expressed as shown in Figure 8b. The output current $I_{ab}$ is given as follows:

$$I_{ab} = \frac{MU_{AB}}{j\omega L_1 L_2} \tag{10}$$

Thus, we can express the transconductance gain, $G_{UI}$, as follows:

$$G_{UI} = \left| \frac{I_{ab}}{U_{AB}} \right| = \frac{M}{j\omega L_1 L_2} \tag{11}$$

For reducing the HFI's VA rating while satisfying the conditions of (8) and (9), the equivalent input impedance can be given as follows:

$$Z_{IN} = \frac{L_1 L_2}{\omega^2 M^2 C_{p1} C_{p2} R_{ac}} \tag{12}$$

The proposed IPT charger can achieve a load-independent CC output under the ZPA condition without constraining the LCT parameters. The output current can be adjusted by changing $L_1$ and $L_2$ to obtain the desired charging current.

### 2.4. Analysis and Implementation of the CC Charging Mode with the ZPA Operation

In this section, the objective is to operate in the CV mode under the ZPA constraint of the proposed hybrid-topology switching IPT system without adding passive compensated components. According to (6) and (7), for the $\Pi-$style network to achieve a CV output, two conditions must be satisfied: $Z_{\Pi 3}$ should tend to infinity and $Z_{\Pi 1} + Z_{\Pi 2} = 0$.

Therefore, we present a new design that uses an ACS to turn off $C_{p2}$ such that $Z_{\Pi 3}$ tends to infinity. Moreover, another ACS is added in $C_{f2}$ to achieve the condition of $Z_{\Pi 1} + Z_{\Pi 2} = 0$. The proposed system's equivalent analytical circuit is presented in Figure 9.

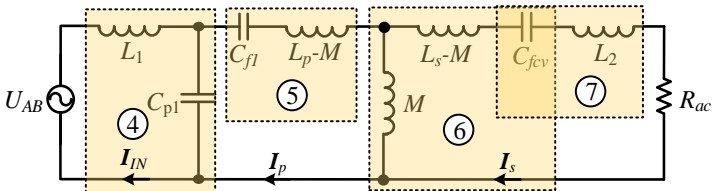

**Figure 9.** The proposed IPT charger's equivalent analytical circuit for the CV mode.

As shown in Figure 9, the analysis method for the primary-side resonant network (④ and ⑤) is identical to that of (① and ②) in Figure 7. $L_1$ and $C_{p1}$ form a 7-style resonant network, which converts the voltage source $U_{AB}$ into a current source. After adjustment by ZPA, it provides energy for the Γ-style resonant network composed of ⑥ and ⑦ on the secondary side. Therefore, the conversion circuit is shown in Figure 10a,b.

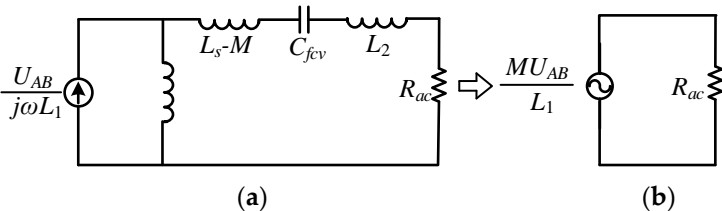

**Figure 10.** The proposed IPT charger's circuit conversion process for the CV mode: (**a**) Circuit before conversion; (**b**) simplified circuit.

For the compensated network on the secondary side to operate in a resonant state, $C_{fcv}$ must resonate $L_s$ and $L_2$ simultaneously. Thus, $C_{fcv}$ can be expressed as follows:

$$C_{fcv} = \frac{1}{\omega^2(L_s + L_2)} \tag{13}$$

To achieve ZVS, it is necessary to realize the ZPA input at the resonance frequency point. In Figure 9, the equivalent input impedance and ZPA of the proposed IPT system after switching topologies can be expressed as:

$$\begin{cases} Z_{IN} = j\omega L_1 + \frac{1}{j\omega C_{p1}} || (j\omega L_p + \frac{1}{j\omega C_{f1}} + Z_R) \\ Q_{IN} = \frac{180°}{\pi} \arctan \frac{\text{Im}(Z_{IN})}{\text{Re}(Z_{IN})} \end{cases} \tag{14}$$

where $Z_r$ represents the reflection impedance, which can be deduced as:

$$Z_R = \frac{\omega^2 M^2}{j\omega L_s + 1/(j\omega C_{fcv}) + j\omega L_2 + R_{ac}} \tag{15}$$

If the conditions of (8) and (13) are satisfied, the equivalent input impedance $Z_{IN}$ can be given as follows:

$$Z_{IN} = \frac{L^2 R_{ac}}{M^2} \tag{16}$$

To obtain the inductive input impedance with the resonant current lagging behind the resonant voltage under the ZVS condition, the resonant frequency can be slightly reduced, or $L_1$ can be set to a value slightly larger than the theoretical value, reducing the inverter's switching loss. Meanwhile, the MOSFETs should have enough dead time to maintain ZVS [33].

Therefore, the rectifier's input voltage, $U_{ab}$, can be seen as a product of the HFI output voltage, $U_{AB}$, multiplied by the ratio of $M$ to $L_1$, which can be expressed as follows:

$$U_{ab} = \frac{U_{AB} M}{L_1} \tag{17}$$

Moreover, the proposed IPT charger's voltage transfer gain, $G_{UU}$, in the CV mode is expressed as:

$$G_{UU} = \frac{U_{ab}}{U_{AB}} = \frac{M}{L_1} \tag{18}$$

As indicated by (18), the proposed IPT charger can realize a CV output and satisfy the charging-voltage requirements of the charging device by adjusting the inductor $L_1$.

## 3. Design and Verification of Proposed Hybrid IPT Charger for CC/CV Modes

### 3.1. The Proposed Hybrid IPT Charger's Parameter Design for CC/CV Modes

According to the foregoing analysis, the proposed IPT charger can easily realize the CC and CV modes under the ZPA constraint. On the charger's primary side, we can modulate $U_{AB}$ using the HFI into a square-wave voltage with a duty cycle of about 0.5 T. In accordance with [34], only the fundamental component is considered. The HFI's output voltage, $U_{AB}$, and output current, $I_{AB}$, and the rectifier's output voltage, $U_b$, and output current, $I_b$, can be expressed as follows:

$$\begin{aligned} U_{AB} &= \frac{2\sqrt{2}}{\pi} U_{DC}, \ I_{AB} = \frac{\pi\sqrt{2}}{4} I_{DC} \\ U_b &= \frac{\pi\sqrt{2}}{4} U_{ab}, \ I_b = \frac{2\sqrt{2}}{\pi} I_{ab} \end{aligned} \tag{19}$$

The battery's equivalent resistance can be expressed as:

$$R_L = \frac{\pi^2}{8} R_{AC} \tag{20}$$

In this study, the compensation parameter symmetrical design method of [22] is adopted. The compensation inductances $L_1$ and $L_2$ and compensation capacitances $C_{f1}$ and $C_{f2}$ are expressed as follows:

$$L_1 = L_2 = \sqrt{\frac{4kU_{AB}}{\omega\pi} \sqrt{\frac{L_p L_s R_{ac}}{P_o}}}. \tag{21}$$

$$C_{f1} = C_{f2} = \frac{1}{\omega^2(L_p - L_1)} = \frac{1}{\omega^2(L_s - L_2)} \tag{22}$$

In the proposed hybrid-compensated IPT system, to avoid adding passive compensated components and reduce the secondary side's volume and cost, $C_{fcc}$ (shown in Figure 5) was designed as a capacitor connected in parallel with $C_{fcv}$. The sum of its capacitance

values is exactly equal to $C_{f2}$ in the DS-LCC topology. $C_{fcc}$ can be determined using (13) and (22), and it can be expressed as:

$$C_{fcc} = \frac{1}{\omega^2(L_s - L_2)} - \frac{1}{\omega^2(L_s + L_2)} \tag{23}$$

Considering the requirements of the charging current and voltage, in addition to the parameter design, the proposed system's detailed compensation parameter design process is expressed in Figure 11. The derived compensated parameters are presented in Table 1.

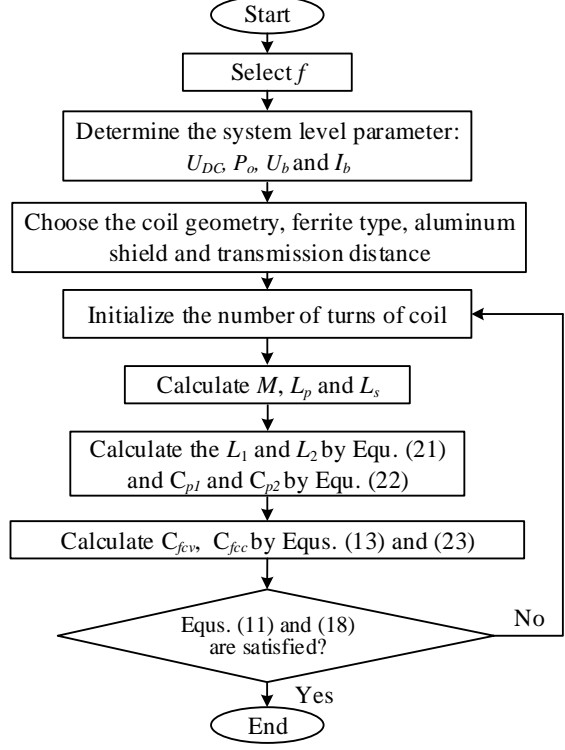

**Figure 11.** The design of the proposed hybrid IPT system's compensation parameters process.

**Table 1.** Designed compensated parameters of the proposed IPT chargers.

| Symbols | Value | Symbols | Value |
|---------|-------|---------|-------|
| $U_{IN}$ | 220 V | $L_1$ | 15.25 μH |
| f | 85 kHz | $L_2$ | 15.15 μH |
| $C_{p1}$ | 229.83 nF | $C_{fcc}$ | 76.025 nF |
| $C_{p2}$ | 229.79 nF | $C_{fcv}$ | 63.045 nF |
| $C_{f1}$ | 145.67 nF | | |

### 3.2. Verification of the Proposed IPT Charger for CC/CV Modes

It is necessary to verify the design's compensation parameters. Therefore, to validate the proposed hybrid IPT charger, we used MATLAB to verify the simulation results as shown in Figure 12a,b. Figure 12 presents the input phase angle, transconductance gain, and voltage gain changes under dynamic loads. As shown, $G_{UI}$ can be kept constant on the premise of ZPA. Similarly, graphs of the voltage gain $G_{UU}$ and input impedance $Z_{IN}$ with respect to the load are presented in Figure 12b. There were smooth voltage and current gains at approximately $f$ = 85 kHz, indicating that the proposed hybrid-compensated IPT system is highly stable.

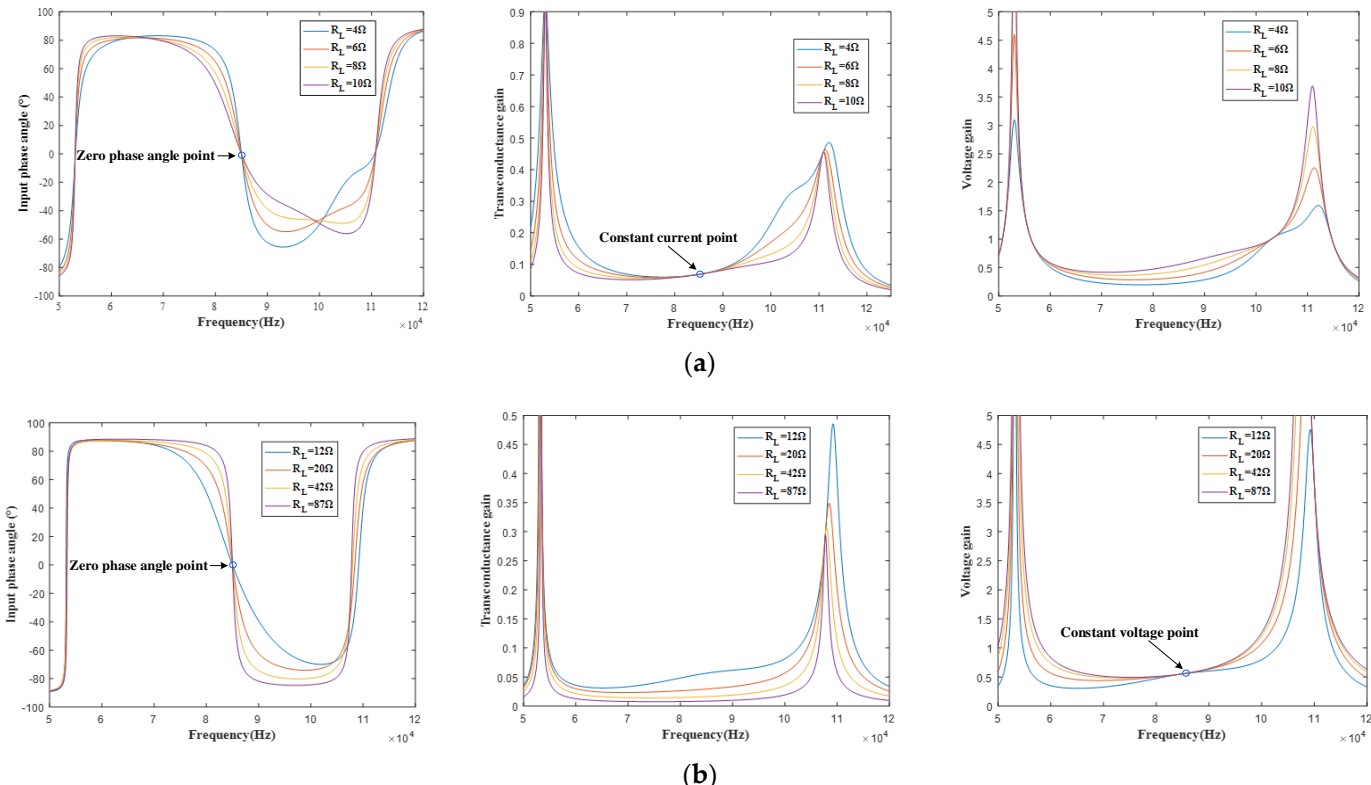

**Figure 12.** The proposed hybrid LCC-LCC topology's input phase angle, voltage transfer gain, and transconductance gain: (**a**) CC mode; (**b**) CV mode.

In this study, ACSs ($S_1$, $S_2$) connected back-to-back by two MOSFETS were adopted to realize the switch from the CC to CV mode. As shown in Figure 13a, the battery charger's ACSs used the same drive signal. Figure 13b shows the logic circuit diagram for the ACSs control drive. $U_b$ represents the charging voltage, and $k_v = U_{REF}/U_b$. As the batteries are charged, because the output voltage of the IPT charger is relatively low, the switches $S_1$ and $S_2$ go into the "on" state and the IPT charger works in the CC output mode. In contrast, as the output voltage reaches the reference voltage $U_{REF}$, $S_1$ and $S_2$ go into the "off" state and the IPT charger operates in the CV output stage.

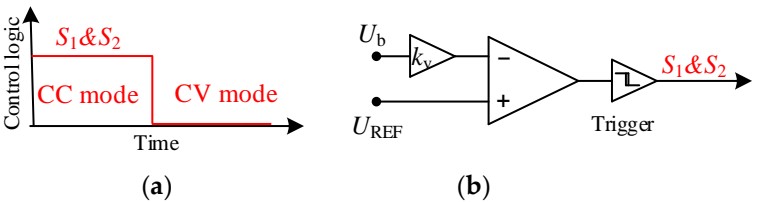

**Figure 13.** (**a**) Control logic for $S_1$ and $S_2$ for the proposed IPT system; (**b**) logic circuit diagram of switching.

## 4. Experiment Verification

To verify the results of the hybrid-topology switching model's theoretical analysis, we established a 1.4 kW experimental prototype based on the circuit diagram, as expressed in Figure 5. The established experimental prototype can provide a 12A CC input and a 120 V CV input to the battery. The experimental parameters are presented in Table 1. An LCR meter (IM3536, HIOKI) was used for measurements. Figure 14 shows the experimental prototype established according to Figure 5. The switching device used in the experiment was a SiC MOSFET (C3M0030090K, WOLFSPEED), which is used for the composition of

HFI and ACSs, controlled by a DSP (TMS320F28335). A Schottky diode (IDW20G120C5B) was used for the rectifier. Dynamic load variations were simulated using Chroma 63200A.

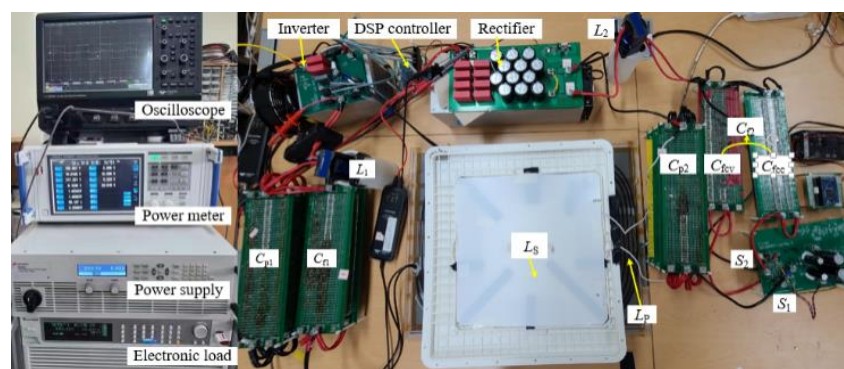

**Figure 14.** Experimental prototype of the proposed IPT battery charger.

In a high-frequency magnetic field, a Litz wire is often adopted to overcome the skin effect, ensuring uniform current distribution through the LCT's cross-section. There is an error of about 1% between the measured and calculation values, and the value setting conformed to the IPT1 standard of SAE J2954, as shown in Table 2. The wireless transmission distance between the transmitting coil and receiving coil was 150 mm. The size of the transmitting coil was 650 × 500 mm, and the size of the receiving coil was 335 × 335 mm.

**Table 2.** Measured parameters of the LCT.

| Symbols | Value | Symbols | Value |
|---------|-------|---------|-------|
| $k$ | 0.206 | M | 8.45 μH |
| $L_p$ | 41.63 μH | $R_{Lp}$ | 36.64 mΩ |
| $L_s$ | 40.41 μH | $R_{Ls}$ | 41.98 mΩ |

Figure 15a presents the curves of the output voltage $U_b$ and output current $I_b$ of the proposed IPT charger with respect to the load. As shown, the charging voltage initially linearly increased with the load, and the charging current remained essentially unchanged, indicating that the IPT system operated in the CC mode. When the output voltage reached $U_{REF}$, the IPT charger switched to the CV charging mode. In contrast to the CC mode, the output voltage remained constant when the IPT charger operated in the CV mode. However, the output current nonlinearly decreased with the increasing load, indicating that a CV output was achieved.

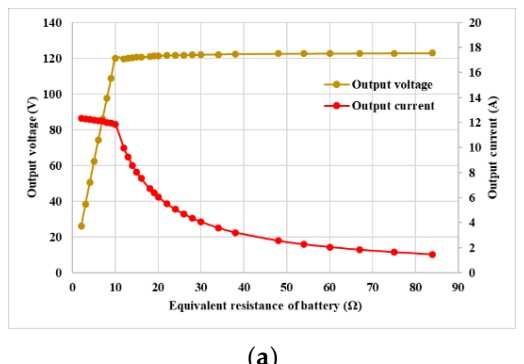

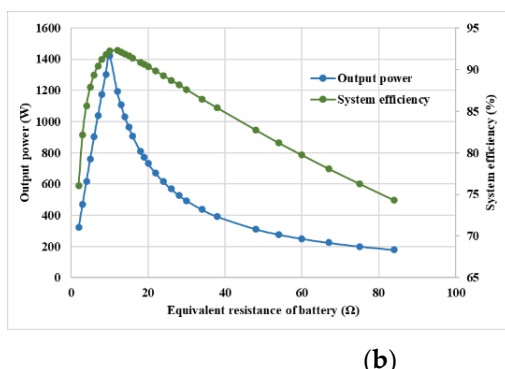

(a)                                                                 (b)

**Figure 15.** The measured battery-charging process under load variation: (**a**) charging voltage and current; (**b**) system efficiency and power.

The IPT charger's efficiency was measured by adopting a power analyzer (HIOKI PW6001) as displayed in Figure 15b. On the left side of the CC/CV modes switching point ($R_L$ = 10.2 Ω), the IPT charger efficiency increased with power. At the mode switching point, the charging efficiency corresponding to the maximum output power of 1.4 kW was 92.2%. On the right side of the mode switching point, the system efficiency increased slightly to maximum efficiency of 92.3%, and then gradually decreased to 75% as the load increased.

To verify the experimental results under the load's dynamic variation, an oscilloscope (Teledyne LeCroy, HDO4034A) was adopted to view the experimental waveforms; this is shown in Figures 16 and 17. Figure 16 presents the experimental waves of the input and output voltages and currents ($U_{AB}$, $I_{AB}$, $U_b$, and $I_b$) when the resistance variations ranged from 6 to 10 Ω in the CC mode.

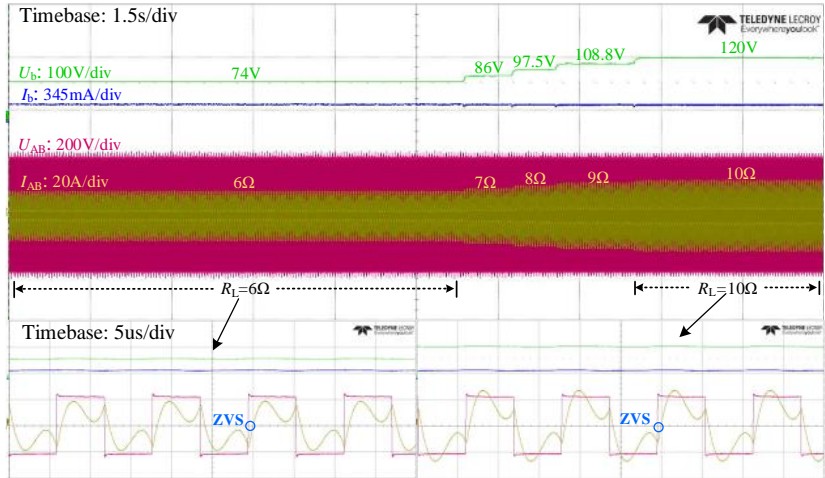

**Figure 16.** Experimental waves of $U_{AB}$, $I_{AB}$, $U_b$, and $I_b$ in the CC mode.

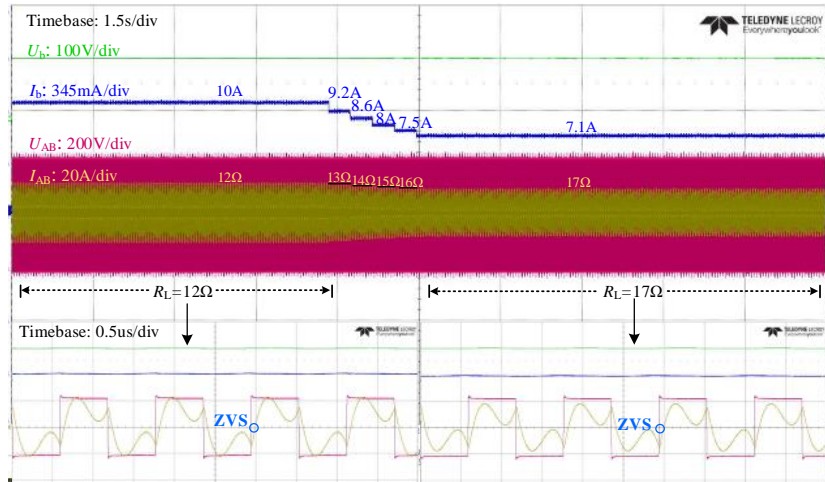

**Figure 17.** Experimental waves of $U_{AB}$, $I_{AB}$, $U_b$, and $I_b$ in the CV mode.

As shown, $U_{AB}$ and $I_b$ remained constant while $I_{AB}$ and $U_b$ changed with respect to the loads. Figure 17 presents the experimental waveforms of the resistance changing from 12 to 17 Ω in the CV mode. As shown, $U_{AB}$ and $U_b$ remained constant, while $I_{AB}$ and $I_b$ dynamically changed with respect to the load. As shown at the bottom of Figures 17 and 18, in the CC/CV charging modes, $I_{AB}$ was in the same phase as $U_{AB}$, where $I_{AB}$ lagged behind $U_{AB}$. This was conducive to achieving ZVS and reducing the HFI loss. When the charging voltage gradually reaches the 120 V battery threshold voltage, it is necessary to observe the transient response process. Figure 18 presents the waveforms of $U_{AB}$, $I_{AB}$, $U_b$, and $I_b$ in the process of turning off $S_1$ and $S_2$. As shown, compared with the case before the

mode switching, the charging voltage and current after the mode switching essentially remained constant, indicating that the CC charging could be smoothly converted into the CV charging mode. Moreover, the charging-mode conversion takes a short time and has almost no effect on the charging process. The experimental results prove the proposed IPT charger's effectiveness.

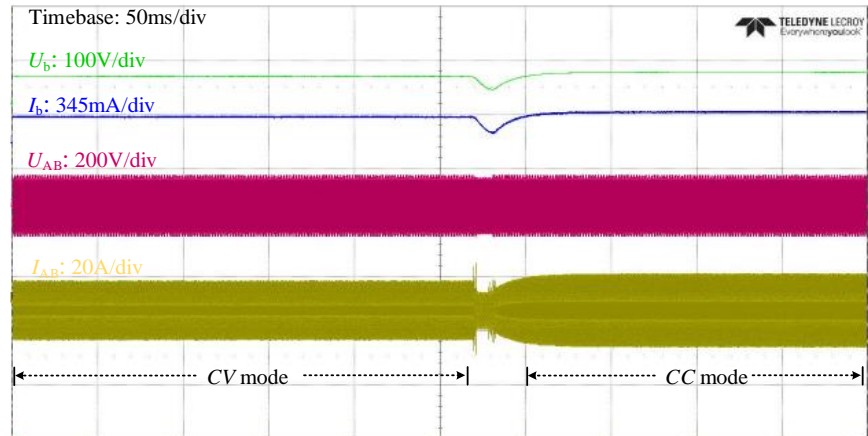

**Figure 18.** Transient waves of $U_{AB}$, $I_{AB}$, $U_b$, and $I_b$ from the CC charging mode to the CV charging mode.

Figure 19 presents the power-loss proportion of each part of the proposed charger. Power loss measured at the switching point ($R_L$ = 10.2 $\Omega$) from the CC to CV mode mainly included HFI, compensation of the inductor and capacitor, LCT, and rectifier losses. Each component's power loss can be deduced through calculating the product of the parasitic resistance and its resonant current. As shown in Figure 19, the losses of the inverter, rectifier, and LCT accounted for the majority of the overall system losses, indicating that their rational design can significantly improve the system's efficiency. Simultaneously, it can be observed that after mode switching, the resonant compensation topology is switched from LCC-LCC to LCC-S, and the current flowing through each component changes, leading to differences in power loss. After switching the compensated topology, the capacitor power loss in the CV mode was reduced by 1.9%. The experimental results for the system efficiency measured by the power analyzer are presented in Figure 20. The highest system efficiency in the CC mode was around 92.2%, and that in the CV mode was around 92.3%. Therefore, the proposed IPT systems are effective in improving the system's efficiency.

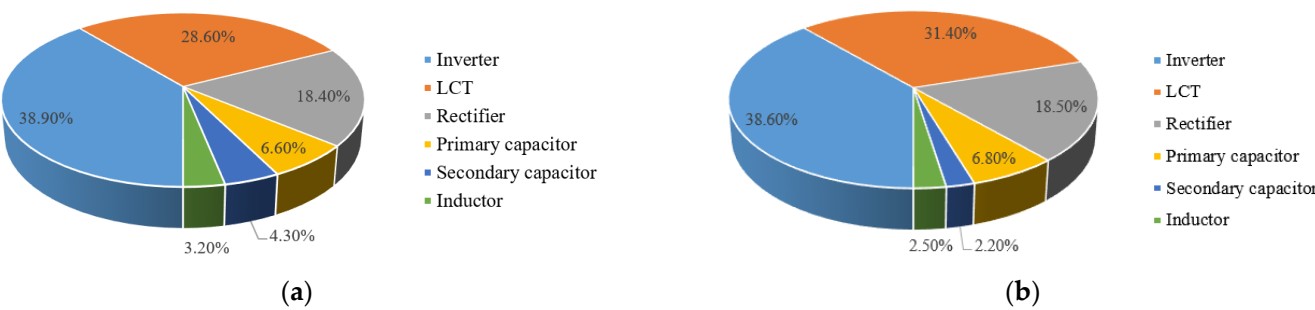

**Figure 19.** The proposed hybrid topology's measured power-loss distribution at $R_L$ = 10.2 $\Omega$: (**a**) CC mode; (**b**) CV mode.

To more intuitively show the proposed system's performance, we conducted a comparative analysis of other methods as shown in Table 3. Firstly, the proposed hybrid compensation IPT system was not limited by LCT, outperforming the methods mentioned in [6,14,19,27,35]. Secondly, secondary-side control was adopted. The configuration of

the wireless communication link can be omitted, which has certain advantages compared to [6,14,26,27,36]. Thirdly, compared with [14], we can observe that the proposed IPT charger avoids the additional DC/DC converter at both sides, effectively reducing the IPT system's cost. Fourthly, the proposed compensation capacitor, $C_{f2}$, on the secondary side of the IPT charger can be fully utilized, which is an advantage that other control methods do not have. Finally, compared with these control methods, the experimental efficiency was relatively high. In short, the proposed control CC/CV IPT charger scheme adopts secondary-side control without compromising design freedom, avoids the configuration of wireless communication link, does not require additional DC/DC converter and compensation capacitors, and has a relatively simple control logic, reducing the proposed IPT charger's cost and complexity.

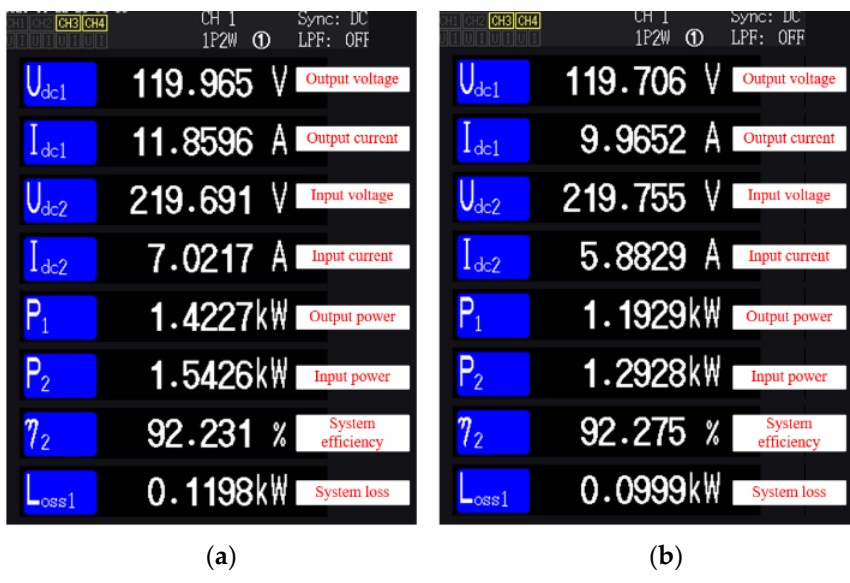

(**a**)   (**b**)

**Figure 20.** Experimental results for the system efficiency measured at the critical point: (**a**) CC mode; (**b**) CV mode.

**Table 3.** Comparison with conventional hybrid compensation topology.

| Proposed in | Ref. [6] | Ref. [14] | Ref. [19] | Ref. [25] | Ref. [26] | Ref. [27] | Ref. [35] | Ref. [36] | This Work |
|---|---|---|---|---|---|---|---|---|---|
| Inductors | 1 | 0 | 1 | 2 | 1 | 1 | 1 | 1 | 2 |
| Capacitors | 4 | 2 | 4 | 5 | 5 | 3 | 3 | 3 | 4 |
| ACSs | 2 | 0 | 2 | 2 | 2 | 1 | 2 | 0 | 2 |
| Number of DC-DC converters | 0 | 1 | 0 | 0 | 0 | 0 | 0 | 0 | 0 |
| Location of ACSs | Receiver | No | Transmitter | Receiver | Both sides | Transmitter | Receiver | No | Receiver |
| Wireless communication | Yes | Yes | Yes | No | Yes | Yes | No | Yes | No |
| Design freedom in CC/CV modes | No/No | No/No | No/Yes | Yes/Yes | Yes/Yes | No/Yes | No/Yes | Yes/Yes | Yes/Yes |
| Control frequency | Fixed | Fixed | Fixed | Fixed | Fixed | Changed | Fixed | Changed | Fixed |
| Max. power | 0.96 kW | 3.25 kW | 3 kW | 0.22 kW | 2.5 kW | 0.2 kW | 2 kW | 3.5 kW | 1.4 kW |
| Peak efficiency | 87.3% | 88.05% | 92.58% | 91.89% | 89.28% | 87% | 92.81% | 97.3% | 92.3% |

## 5. Conclusions

This study aimed to develop an IPT charger capable of charging batteries in the CC and CV modes based on an analysis of the Π-style network, in addition to effectively utilizing existing configuration compensation parameters and hybrid-topology switching. This paper proposed an IPT charger with hybrid-topology switching that does not require real-time communication links. Our main contributions are summarized as follows:

(1) Only two ACSs had to be added on the secondary side. No additional compensation devices were needed and increases in the volume and weight were avoided. This satisfied the requirement of secondary side portability in the IPT system design.

(2) The ZPA operation in the CC/CV modes can be easily realized, enhancing the proposed IPT charger's efficiency.

(3) Because the same driving signal was used to control the ACSs, the control strategy was relatively simple.

(4) Secondary-side control was adopted. The configuration of the wireless communication links can be omitted to prevent the electromagnetic field from causing interference to the wireless communication links, which ensures the proposed charger's robustness and reliability.

An experimental prototype of a 1.4 kW IPT charger was constructed to verify the performance of the proposed IPT systems. Experimental results indicated that it could achieve efficient CC and CV outputs regardless of load variation.

**Author Contributions:** Conceptualization, G.L., C.-S.S. and D.-H.K.; data curation, G.L.; formal analysis, G.L.; funding acquisition, D.-H.K.; investigation, C.-H.J. and D.-H.K.; project administration, S.J. and D.-H.K.; writing—original draft, G.L.; writing—review and editing, D.-H.K. All authors have read and agreed to the published version of the manuscript.

**Funding:** This work was supported by the Technology Innovation Program (or Industrial Strategic Technology Development Program-Automobile industry technology development) (20018829, Vehicle Underbody Charging Type Automatic Charging and Management/Control System Development) funded By the Ministry of Trade, Industry & Energy (MOTIE, Korea).

**Institutional Review Board Statement:** Not applicable.

**Conflicts of Interest:** The authors declare no conflict of interest.

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
