# Peer review of "A Load-Independent Current/Voltage IPT Charger with Secondary Side-Controlled Hybrid-Compensated Topology for Electric Vehicles"

_applsci, doi:10.3390/app122110899_

Round 1
Reviewer 1 Report
This paper proposes a IPT charger design capable of providing a load-independent output, a secondary side-controlled hybrid-compensated topology for to charge the battery with a constant current/voltage output. This subject is properly presented with a good scientific soundness.
Reviewer 2 Report
A secondary-side controlled hybrid-compensated IPT was proposed in this paper. By adding AC switches, the proposed IPT can operate normally. However, there are several problems in this paper:
1. Fig. 8 is wrong.
2. The author is supposed to analyze how the circuit given in Fig. 9 is reduced to Fig. 10(a).
3. A comparator is used to determine the state switch from CC to CV. If the output voltage is very close to the reference voltage, there will be repeated state switching. How to solve this problem?
4. In actual application, as shown in Fig. 2, the battery's equivalent resistance exhibits a nolinear trend. But the V-R, I-R curves were given in the experiment results instead of the V-t, I-t cuives.
Reviewer 3 Report
It requires significant improvement. Writing can be improved, no point to capitalized the first letter, e.g. Power Factor Correction (PFC). It is strongly suggested to check the requirement of basic theories including figures from existing knowledge. If needed, relevant references should be cited. Quality of most of figures are not acceptable, relevant figures should be replaced with more quality ones. Contributions of the paper should be more clear and well supported. It is suggested to add a photograph of the test platform. Not well clear which section of the system is tested practically.
Round 2
Reviewer 3 Report
This version can be accepted.